# Reconfigurable all-dielectric metalens with diffraction-limited performance

Mikhail Y. Shalaginov [1,7], Sensong An [2,7], Yifei Zhang [1], Fan Yang[1], Peter Su[1], Vladimir Liberman[3], Jeffrey B. Chou[3], Christopher M. Roberts [3], Myungkoo Kang[4], Carlos Rios [1], Qingyang Du[1], Clayton Fowler[2], Anuradha Agarwal[1,5], Kathleen A. Richardson[4], Clara Rivero-Baleine[6], Hualiang Zhang[2✉], Juejun Hu[1✉] & Tian Gu[1,5✉]

Active metasurfaces, whose optical properties can be modulated post-fabrication, have emerged as an intensively explored field in recent years. The efforts to date, however, still face major performance limitations in tuning range, optical quality, and efficiency, especially for non-mechanical actuation mechanisms. In this paper, we introduce an active metasurface platform combining phase tuning in the full $2\pi$ range and diffraction-limited performance using an all-dielectric, low-loss architecture based on optical phase change materials (O-PCMs). We present a generic design principle enabling binary switching of metasurfaces between arbitrary phase profiles and propose a new figure-of-merit (FOM) tailored for reconfigurable meta-optics. We implement the approach to realize a high-performance varifocal metalens operating at 5.2 μm wavelength. The reconfigurable metalens features a record large switching contrast ratio of 29.5 dB. We further validate aberration-free and multi-depth imaging using the metalens, which represents a key experimental demonstration of a non-mechanical tunable metalens with diffraction-limited performance.

[1] Department of Materials Science & Engineering, Massachusetts Institute of Technology, Cambridge, MA, USA. [2] Department of Electrical & Computer Engineering, University of Massachusetts Lowell, Lowell, MA, USA. [3] Lincoln Laboratory, Massachusetts Institute of Technology, Lexington, MA, USA. [4] The College of Optics & Photonics, Department of Materials Science and Engineering, University of Central Florida, Orlando, FL, USA. [5] Materials Research Laboratory, Massachusetts Institute of Technology, Cambridge, MA, USA. [6] Missiles and Fire Control, Lockheed Martin Corporation, Orlando, FL, USA. [7] These authors contributed equally: Mikhail Y. Shalaginov, Sensong An. ✉email: hualiang_zhang@uml.edu; hujuejun@mit.edu; gutian@mit.edu

The ability to reconfigure an optical component, thereby tuning its optical response to meet diverse application demands at will, has been a long-sought goal for optical engineers. Traditionally, such dynamic reconfigurability often requires bulky mechanical moving parts, for example in a zoom lens. The approach, however, usually comes with the price of increased system size and complexity. Unlike conventional optics that rely on geometric curvature to mold the propagation phase of light, metasurfaces afford on-demand control of an optical wavefront using subwavelength antenna arrays patterned via standard planar microfabrication technologies[1–6]. In addition to their potential size, weight, power, and cost (SWaP-C) benefits, they also present a versatile suite of solutions to realizing reconfigurable optical systems, leveraging so-called "active metasurfaces", whose optical responses can be dynamically tuned.

Over the past few years, active metasurfaces have been investigated intensively[7–14]. Mechanical deformation or displacement of metasurfaces is an effective method for tuning metasurface devices or adaptively correcting optical aberrations[15–19]. On the other hand, non-mechanical actuation methods, which allow direct modulation of optical properties of meta-atoms, can offer significant advantages in terms of speed, power consumption, reliability, as well as design flexibility. A variety of tuning mechanisms such as free carrier[20], thermo-optic[21], electro-refractive[22], and all-optical[23] effects have been harnessed to create active metasurface devices. However, these effects are either relatively weak (e.g., thermo-optic, electro-refractive, and all-optical effects) or incur excessive optical loss (e.g., free carrier injection). Consequently, the tuning range and optical efficiency of these active metasurfaces are often limited.

Phase change and phase transition materials (exemplified by chalcogenide compounds and correlated oxides such as $VO_2$, respectively) offer another promising route for realizing active metasurfaces[13,24–26]. The extremely large refractive index contrast associated with material phase transformation (e.g., $\Delta n > 1$) uniquely empowers metasurface devices with ultra-wide tuning ranges. Many studies have achieved amplitude or spectral tailoring of light via metastructures made of these materials[27–37]. Tunable optical phase or wavefront control, which is essential for realizing multifunctional meta-optical components, such as, metalenses and beam-steering devices, has also been demonstrated[38–41]. However, that meta-optical devices had relatively low efficiencies, and their phase precision, a key metric that dictates optical quality of metasurface devices, has not been quantified. Moreover, the designs often suffer from significant crosstalk between the optical states, which causes ghosting across the variable states and severe image quality degradation in imaging applications. As a result, it is not clear yet whether dynamic meta-optical devices can possibly attain diffraction-limited, low-crosstalk performances rivaling their traditional bulky refractive counterparts.

Besides experimental implementation, the design of wavefront-shaping devices based on active metasurfaces also poses a largely unexplored challenge. The presence of two or more optical states vastly increases the complexity of design targets. In addition, modulating the optical properties of meta-atoms in general concurrently modifies their phase and amplitude responses, both of which impact the device performance in its different optical states. Optimization of dynamic meta-optical devices, therefore, requires a computationally efficient design composition and validation approach to generate meta-atom libraries that allow down selection of optimal meta-atom geometries, which yield the desired optical performance at each state.

In this paper, we present a generic design methodology, enabling switching of metasurface devices to realize arbitrary phase profiles. A new figure-of-merit (FOM) suited for tunable

meta-optics is developed to facilitate efficient and accurate metasurface performance prediction without resorting to computationally intensive full-system simulations. The design framework is validated through demonstration of high-performance varifocal metalens. The concept of a varifocal lens based on phase-change materials was first elegantly implemented in the pioneering work by Yin et al.[41]. Their design relied on two groups of plasmonic antennae sharing the same lens aperture on top of a blanket phase-change material film, each of which responded to incident light at either the amorphous or crystalline state of the film. The shared-aperture layout and the use of metallic meta-atoms limited the focusing efficiencies to 5% and 10% in the two states. Focal spot quality of the lens was also not reported. Our device instead builds on all-dielectric meta-atom structures optimized via design methodology to simultaneously minimize phase error (thereby suppressing crosstalk) and boost optical efficiency. The design FOM allows computationally efficient synthesis of the active metasurface without performing simulations for each optical system during the optimization process, and thus it is scalable to designs with increased complexity and functionalities. We have further experimentally demonstrated diffraction-limited imaging free of aberration and crosstalk at both states of the metalens, proving that active metasurface optics based on O-PCM technologies can indeed attain a high level of optical quality matching that of their conventional bulk counterparts while taking full advantage of their flat optical architecture.

## Results

**On-demand composition of bi-state meta-optical devices: concept and design methodology.** We selected $Ge_2Sb_2Se_4Te_1$ (GSST) as a non-volatile O-PCM to construct the metasurface operating at the wavelength $\lambda_0 = 5.2 \, \mu m$. Compared to the classical $Ge_2Sb_2Te_5$ (GST) phase-change alloy, GSST offers exceptionally broadband transparency in the infrared spectral regime for both its amorphous and crystalline phases, a feature critical to optical loss reduction, while maintaining a large refractive index contrast between the two states[42,43]. The metasurface consists of patterned, isolated GSST Huygens' meta-atoms sitting on a $CaF_2$ substrate (Fig. 1). The Huygens-type meta-atom design features an ultra-thin, deep subwavelength profile ($<\lambda_0/5$), which facilitates a simple one-step etch fabrication process[44–47]. Although here we use a bi-state varifocal metalens as our proof-of-concept demonstration, our device architecture and design approach are generic and applicable to active metasurfaces switchable between arbitrary phase profiles. The design can also be readily generalized to active metasurfaces supporting more than two optical states, for instance, leveraging intermediate states in O-PCMs[48,49].

The design procedure of the reconfigurable metalens with a dimension of $1.5 \times 1.5 \, mm^2$ is illustrated in Fig. 2. The design process starts by defining the target phase maps in the two optical

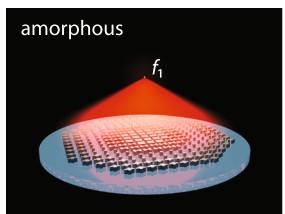
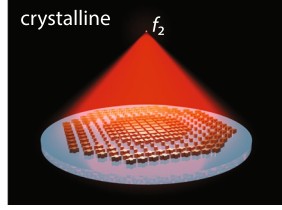

**Fig. 1 Artistic rendering of a reconfigurable varifocal metalens.** Incident light is focused on the first focal plane ($f_1 = 1.5 \, mm$) when the meta-atoms are in the amorphous state and the second focal plane ($f_2 = 2.0 \, mm$) in the crystalline state.

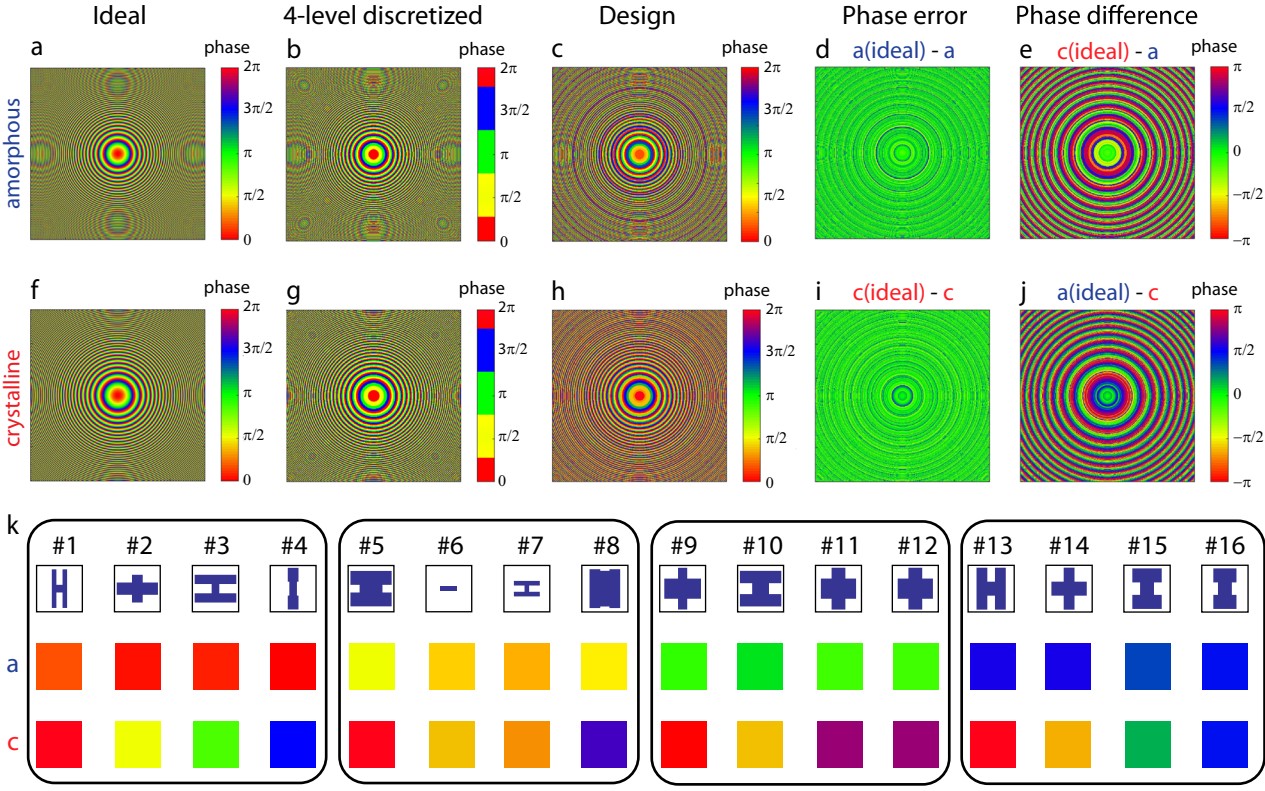

**Fig. 2 Phase maps of the metalens in amorphous and crystalline states. a**, **f** Ideal target phase profiles with continuous phase distribution; **b**, **g** four-level discretized phase profiles; and **c**, **h** final design taking into account phase responses of the meta-atoms. Difference between the ideal and final design phase maps at the **d**, **i** primary and **e**, **j** phantom focal planes. **k** 16 meta-atoms selected to construct the reconfigurable metalens. Colors correspond to the phase values shown in **c**, **h**.

states. For the varifocal metalens under consideration, two hyperbolic phase profiles (with $2\pi$ phase wraps) yielding focal lengths of $f_1 = 1.5$ mm (amorphous, a-state) and $f_2 = 2$ mm (crystalline, c-state) are plotted in Figs. 2a and 2f, respectively. The design corresponds to numerical aperture (NA) values of 0.45 and 0.35 in the amorphous and crystalline states, respectively. We then choose to discretize the continuous 0 to $2\pi$ phase profiles into $m = 4$ phase levels, i.e., 0, $\pi/2$, $\pi$, and $3\pi/2$ (Figs. 2b, g). To enable switching between two arbitrary phase profiles with four discrete phase levels, a total of $m^2 = 16$ meta-atom designs are needed, each of which provides a distinct combination of two of the four discrete phase values during the phase transition. An ideal meta-atom design must minimize phase error while maximizing optical efficiency at both states. Realistic designs however often face trade-offs between phase error and efficiency given the inherent complexity associated with the bi-state design targets, as detailed further.

To obtain the 16 optimal meta-atom designs, a pool of Huygens' meta-atoms with various regular geometries, such as 'I', 'H', and "+" shapes, were first generated by sweeping the geometric parameters in a full-wave electromagnetic solver (Supplementary Note 1), and then grouped according to the four-phase levels and phase variances between the two states. Different sub-groups of meta-atoms were then mapped onto the evenly-discretized metasurface phase profiles. Using the generated phase/amplitude masks and following generalized diffraction efficiency calculation of multi-level diffractive optical elements[50], we develop a new performance FOM suitable for evaluating and optimizing meta-atom designs without resorting to full-scale system simulations. Derivation of the FOM is detailed in Supplementary Note 3. The 16 meta-atoms

were selected from the design pool based on the following FOMs:

$$\mathrm{FOM}_{1,a} = T_{avg,a} \cdot \left( \frac{\sin\left(2\left\langle \left| \phi_{meta,a} - \phi_{target,a} \right| \right\rangle\right)}{2\left\langle \left| \phi_{meta,a} - \phi_{target,a} \right| \right\rangle} \right)^2 \quad (1)$$

$$\mathrm{FOM}_{2,c} = T_{avg,c} \cdot \left( \frac{\sin\left(2\left\langle \left| \phi_{meta,c} - \phi_{target,c} \right| \right\rangle\right)}{2\left\langle \left| \phi_{meta,c} - \phi_{target,c} \right| \right\rangle} \right)^2 \quad (2)$$

$$\mathrm{FOM}_{eff} = \sqrt{\mathrm{FOM}_{1,a} \cdot \mathrm{FOM}_{2,c}} \quad (3)$$

where $\mathrm{FOM}_{1,a}$ and $\mathrm{FOM}_{2,c}$ correlate with the metasurface performances on focal plane $f_1$ in the amorphous state and focal plane $f_2$ in the crystalline state, respectively. $T_{avg,a(c)}$, $\phi_{target,a(c)}$, and $\phi_{meta,a(c)}$ are the average meta-atom transmittance, target phase values and simulated actual phase values in the amorphous (crystalline) state. Maximization of $\mathrm{FOM}_{eff}$ ensures good focal spot quality and focusing efficiency at both optical states, which provides quantitative evaluation of the trade-offs between efficiency and phase error. This in turn enables the synthesis of a metasurface with the best meta-atom structures without performing full-scale simulations of the entire optical system. Implementation of the aforementioned FOM evaluation method can be further extended from the metasurface level to the meta-atom level before constructing a specific metasurface design, by applying weighting factors to different meta-atom geometries according to the metasurface phase map.

In imaging applications involving reconfigurable metalenses, crosstalk is another extremely important metric, since crosstalk results in ghost image formation, which severely degrades image quality. The FOM defined above can be revised to further take into account crosstalk between the two states, which is characterized by the switching contrast ratio CR:

$$\mathrm{CR} = 10 \log_{10}\left(\frac{P_{1,a}}{P_{2,a}} \cdot \frac{P_{2,c}}{P_{1,c}}\right) \quad \text{(in dB)} \tag{4}$$

where $P_{1(2),a(c)}$ denotes the focused optical power (defined as the power confined within a radius of $5\lambda_0$) at focal spot 1 (2) at the amorphous (crystalline) state. As the ideal phase profiles are already defined in Figs. 2a, f, the focused power at the "phantom" focal spot (i.e., focal spot 2 in the a-state and focal spot 1 in the c-state) is solely determined by the "difference" in the two phase profiles. In practice, the CR can be compromised by incomplete switching of the meta-atoms and is thus also an essential measure of the metalens' optical quality. For the cases of diffractive optical elements (DOEs) or metasurfaces (the latter of which are sometimes regarded as multi-level DOEs with a subwavelength array), phase deviations mostly originate from random errors due to the phase sampling process, as compared with continuous and systematic wavefront distortions, which are typically encountered in refractive bulk optics. Consequently, the RMS phase errors of such devices mostly contribute to scattered loss or crosstalk between optical states, as analyzed for multi-level DOEs in ref. [50]. We note that FOMs defined in Eqs. 1 and 2 scale directly with diffraction efficiency, or specifically in the case of a metalens, the focusing efficiency on a particular focal plane in a particular state. The equations can therefore be equally applied to correlate light intensities at the phantom focal spots with the metasurface design. Thus, a FOM taking CR into account can be developed based on Eqs. (1, 2, and 4):

$$\mathrm{FOM}_{2,a} = T_{avg,a} \cdot \left(\frac{\sin\left(2\left\langle\left|\phi_{meta,a} - \phi_{target,c}\right|\right\rangle\right)}{2\left\langle\left|\phi_{meta,a} - \phi_{target,c}\right|\right\rangle}\right)^2 \tag{5}$$

$$\mathrm{FOM}_{1,c} = T_{avg,c} \cdot \left(\frac{\sin\left(2\left\langle\left|\phi_{meta,c} - \phi_{target,a}\right|\right\rangle\right)}{2\left\langle\left|\phi_{meta,c} - \phi_{target,a}\right|\right\rangle}\right)^2 \tag{6}$$

$$\mathrm{FOM}_{CR} = \frac{\mathrm{FOM}_{1,a}}{\mathrm{FOM}_{2,a}} \cdot \frac{\mathrm{FOM}_{2,c}}{\mathrm{FOM}_{1,c}} \tag{7}$$

where $\mathrm{FOM}_{2,a}$ and $\mathrm{FOM}_{1,c}$ relate to the metasurface's "ghosting" performance on focal plane $f_2$ in the amorphous state and focal

plane $f_1$ in the crystalline state, respectively, and are proportional to the optical efficiencies of the ghost images in both states.

The FOMs were evaluated for metalens design variants assembled from meta-atoms within the pool. Specifically, phase masks with phase and amplitude responses of the meta-atoms simulated from full-wave models were employed to simulate the metasurface performance using the Kirchhoff diffraction integral, a physically rigorous form of the Huygens-Fresnel principle[51]. The diffraction integral allows computationally efficient validation of the metalens performance not constrained by the large lens size (1.5 mm × 1.5 mm square aperture). In all, 16 meta-atom geometries which yield the maximum FOM were chosen to assemble the final metalens design (as shown in Fig. 2k). More details on the selected meta-atom shapes and optical responses are available in Supplementary Note 2. The phase deviations of the final design (Figs. 2c, h) from the ideal phase profile are shown in Figs. 2d, i with a negligible average phase error of <0.013 $\lambda_0$ for both states, root-mean-square (RMS) errors of 0.11 $\lambda_0$ and 0.17 $\lambda_0$; average meta-atom transmittance was 67% and 71%, in the amorphous and crystalline states, respectively. In contrast, the phase errors on the phantom focal planes are significantly larger (Figs. 2e, j). Simulations using the diffraction integral model incorporating the phase/amplitude masks yield Strehl ratios close to unity (>0.99) for both states and focusing efficiencies of 39.5% and 25.4% in the amorphous and crystalline states, respectively. The optical efficiencies are mainly restricted by the small number of phase discretization levels ($m = 4$) and limited transmittance of the meta-atoms. The simulations further yield power ratios of $\sim P_{1,a}/P_{2,a} = 453$ and $P_{2,c}/P_{1,c} = 36$ in amorphous and crystalline states, respectively, corresponding to a theoretical CR of 42.1 dB. In addition, we analyzed the metalens' diffraction-limited bandwidths (i.e., wavelength range over which Strehl ratios exceed 0.8), which are ~80 nm and 100 nm for amorphous and crystalline states, respectively. The operational bandwidths are in good agreement with the dispersion behavior of an ideal flat lens of the same configuration (Supplementary Note 5).

**Metalens fabrication and characterization.** The metalens was patterned in thermally evaporated GSST films on a CaF₂ substrate using electron beam lithography (EBL) and standard plasma etching. More details of the metalens fabrication are furnished in the Methods section. Figure 3 presents scanning electron microscopy images of the fabricated metasurfaces. The meta-atoms show negligible surface roughness, almost vertical sidewalls with a sidewall angle >85°, and excellent pattern fidelity consistent with our design.

The metalens was characterized using an external cavity tunable quantum cascade laser (QCL) emitting linearly polarized

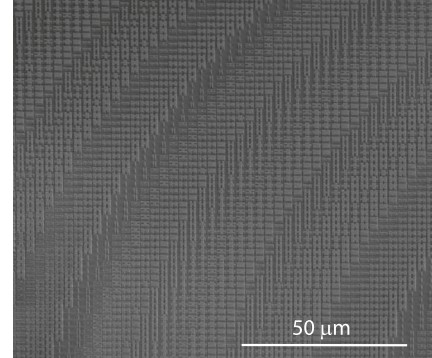
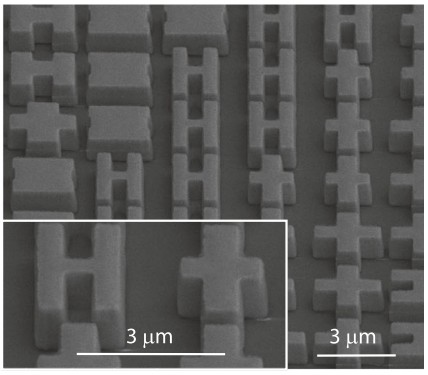

**Fig. 3 SEM scans of the metalens.** The images show the GSST meta-atoms with vertical sidewalls and excellent pattern fidelity.

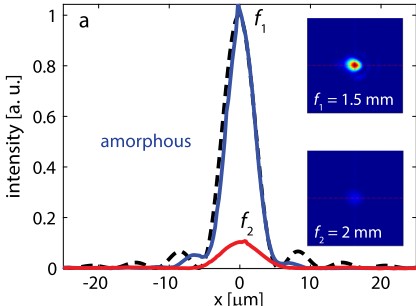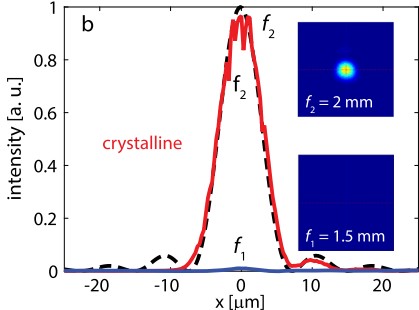

**Fig. 4 Optical characterization.** Focal spot profiles for the metalens in two states: **a** amorphous and **b** crystalline. Each plot contains the focal spot intensity distributions for the $f_1 = 1.5$ mm and $f_2 = 2$ mm focal planes. All the focal spots are diffraction-limited. The focal spots produced by ideal, aberration-free lenses of the same numerical aperture are marked with black dashed-curves. The insets show the 2-D images of the focal spots: $f_1 = 1.5$ mm and $f_2 = 2$ mm. Power contrast ratios are 10:1 and 90:1 for the a- and c-states, respectively.

light at 5.2 μm wavelength. The collimated laser beam was focused by the metalens and images of the focal spots were first magnified with a double-lens microscope assembly (with a calibrated magnification of 120) and then recorded by a liquid nitrogen cooled InSb focal plane array (FPA) camera on the two focal planes ($f_1 = 1.5$ mm and $f_2 = 2$ mm). The focal spot images are shown in Fig. 4 insets and the main panels in Fig. 4 plot the optical intensity profiles across the center planes of the focal spots along with those of ideal aberration-free lenses of the same NAs. The metalens features high Strehl ratios of >0.99 and 0.97 in the amorphous and crystalline states, respectively, implying that the lens operates in the diffraction-limited regime at both states. We further experimentally measured the focused power ratios between the true and phantom focal spots, yielding $P_{1,a}/P_{2,a} = 10$ and $P_{2,c}/P_{1,c} = 90$. The result corresponds to a large CR of 29.5 dB, the highest reported value to date in active metasurface devices (Supplementary Table 3).

Focusing efficiency of the metalens was quantified following our previously established measurement protocols[52]. Focusing efficiencies of 23.7% and 21.6% were measured for the amorphous and crystalline states, respectively. The difference between the experimental results and theoretical predictions are primarily owing to meta-atom geometry and refractive index deviations in the fabricated device. However, the demonstrated performance still represents major improvements over prior state-of-the-art in varifocal metalens (Supplementary Table 1).

Finally, we demonstrated high-resolution, low-crosstalk imaging using our reconfigurable metalens. Standard USAF-1951 resolution charts in the form of Sn patterns fabricated on $CaF_2$ disks were used as the imaging objects. The imaging object comprises one or two resolution charts coinciding with the two focal planes ($f_1 = 1.5$ mm and $f_2 = 2$ mm), which are flood-illuminated from the backside using the QCL. The metalens was used as an objective to project the resolution target images onto the camera. Figure 5a shows four images of the resolution charts captured using the setup when only a single resolution target was placed at one of the focal planes. The lens produced clearly resolved images of the USAF 6.2 (half period 7 μm) and USAF 5.6 (half period 8.8 μm) patterns when the lens was in amorphous and crystalline states, respectively. This result agrees well with theoretical resolution limits of 7 μm and 9 μm in the two states, suggesting that our metalens can indeed achieve diffraction-limited imaging performance. In contrast, no image was observed when the resolution target was placed at the phantom focal plane.

We further show that the metalens can be used for imaging multi-depth objects with minimal crosstalk. In the test, two resolution targets were each positioned at one focal plane with 45° relative in-plane rotation with respect to the other target. At each

optical state of the metalens, only one resolution target aligning with the focal plane was clearly imaged with no sign of ghost image resulting from the other target (Fig. 5b). These results prove that the reconfigurable metalens is capable of diffraction-limited imaging free of optical aberrations and crosstalk across overlapping objects at different depths.

## Discussion

Our work demonstrates that judiciously engineered active meta-surfaces can achieve high optical quality in the diffraction-limited regime rivaling the performance of traditional aspheric refractive optics. The high-performance meta-optics as well as the efficient design approach will open up many exciting applications involving reconfigurable or adaptive optics. For instance, the varifocal metalens constitutes a key building block for a parfocal lens (a true zoom lens that stays in focus while changing magnification) widely used in cameras, microscopes, telescopes, and video recorders. Conventional parfocal zoom lenses necessarily involve multiple mechanically moving elements required for aberration compensation while tuning the magnification, which severely compromise the size, complexity, ruggedness, and often image quality. In contrast, our varifocal metalens enables a drastically simplified step-zoom parfocal lens design consisting of only two phase-change metasurfaces patterned on the top and bottom surfaces of a single flat substrate, while maintaining diffraction-limited imaging performance. Besides imaging, the active meta-surface can potentially also enable other applications such as beam steering, adaptive optics, and optical spectroscopy[53].

Switching from the amorphous to the crystalline phase was accomplished via furnace annealing in our present prototype. Practical deployment of the reversible reconfigurable metasurface will necessarily involve electrical switching of O-PCMs. We have recently demonstrated highly consistent electrothermal switching of GSST over 1000 phase transition cycles using on-chip metal micro-heaters[42]. In addition, reversible switching of GSST and other phase-change materials using transparent graphene[54], indium-tin oxide[55,56], and doped Si[57] heaters have also been validated. In this regard, the use of GSST rather than the classical GST alloy uniquely benefits from not only GSST's low optical attenuation but also its improved amorphous phase stability. GST boasts a short crystallization time in the nanosecond regime[58], which is useful for ultrafast switching but at the same time also limits the critical thickness amenable to fully reversible switching to <100 nm. In comparison, whereas the detailed crystallization kinetics of GSST has not yet been quantified[42], its crystallization time is likely in the order of microseconds. This much longer crystallization time permits reversible switching of GSST films

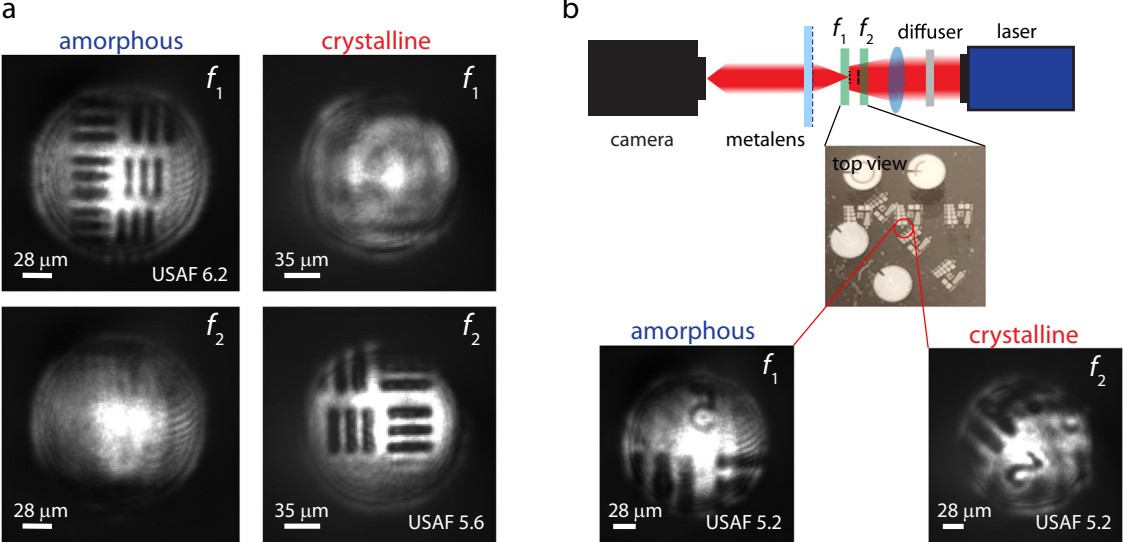

**Fig. 5 Imaging using the GSST varifocal metalens. a** Well-resolved lines of USAF-1951 resolution charts: the patterns have half periods close to the Rayleigh resolution limits of 7 μm and 9 μm in the a-state ($f_1$) and c-state ($f_2$), respectively. **b** Schematic of the setup for imaging multi-depth targets. Top-view photograph of the target consisting of two patterned samples overlapped at an angle of 45°. Camera images of the dual-depth target acquired by stationary metalens in a- and c-states.

with thicknesses exceeding 1 μm, presenting a critical benefit for their photonic applications. Indeed, we have recently reported an electrically reconfigurable metasurface based on O-PCMs at the 1550 nm telecommunication band, where the entire meta-atoms (250 nm in thickness) are made of GSST. The ensuing large optical modal overlap with the O-PCM enables spectral tuning of resonances across a record broad half-octave band[59]. These advances define a clear path towards practical implementation of the active metasurface design with integrated transparent heaters. It is worth mentioning that our general reconfigurable metasurface design principle can be also implemented with other O-PCMs, such as $Sb_2S_3$[60], $Sb_2Se_3$[61], and $Ge_3Sb_2Te_6$[62].

Finally, even though our metalens already claims exceptional optical quality, our generic design principle points to several future improvements, which can further enhance lens performance and design versatility. Our present metalens uses four discrete phase levels, which imposes ~20% efficiency loss due to discretization phase errors[63]. The conventional searching method based on parameter sweeping limits the size of the accessible unit cell library in practice. Increasing the number of phase discretization levels $m$ contributes to mitigating phase errors and increasing focusing efficiency. One can also scale the design approach to three or more arbitrary optical states taking advantage of intermediate states and the large index contrast afforded by O-PCMs[48,49]. In general, an active metasurface with $j$ optical states ($j \geq 2$) each characterized by $m$ phase levels demands a minimum of $m^j$ distinct meta-atoms. The design problem, whose complexity escalates rapidly with increasing $m$ and $j$, is best handled with deep learning based meta-atom design algorithms[64,65] and will be the subject of a follow-up paper.

In conclusion, we propose a non-mechanical active metasurface design to realize binary or multi-configuration switching between arbitrary optical states. We validated the design principle by fabricating a varifocal metalens using low-loss O-PCM GSST, and demonstrated aberration and crosstalk-free imaging. The work proves that non-mechanical active metasurfaces can achieve optical quality on par with conventional precision bulk optics involving mechanical moving parts, thereby pointing to a cohort of exciting applications fully unleashing the SWaP-C benefits of active metasurface optics in imaging, sensing, display, and optical ranging.

## Methods

**Metasurface fabrication.** GSST films of nominally 1 μm thickness were deposited onto a double-side polished CaF₂ (111) substrate (MTI Corp.) by thermal co-evaporation in a custom-made system (PVD Products Inc.)[66]. The desired film stoichiometry was achieved by controlling the ratio of evaporation rates of two isolated targets of $Ge_2Sb_2Te_5$ and $Ge_2Sb_2Se_5$. The deposition rates were kept at 4.3 Å/s ($Ge_2Sb_2Te_5$) and 12 Å/s ($Ge_2Sb_2Se_5$) with a base pressure of $2.8 \times 10^{-6}$ Torr and a sample holder rotation speed of 6 rpm. The substrate was held near room temperature throughout the film deposition process. Thickness of the film was measured with a stylus profilometer (Bruker DXT) to be 1.10 μm (a-state) and 1.07 μm (c-state), indicating 3% volumetric contraction during crystallization similar to other phase-change materials[67,68]. The film was patterned via EBL on an Elionix ELS-F125 system followed by reactive ion etching (Plasmatherm, Shuttle-lock System VII SLR-770/734). The electron beam writing was carried out on an 800-nm-thick layer of ZEP520A resist, which was spin coated on top of the GSST film at 2000 rpm for 1 min and then baked at 180 °C for 1 min. Before resist coating, the sample surface was treated with standard oxygen plasma cleaning to improve resist adhesion. To prevent charging effects during the electron beam writing process, the photoresist was covered with a water-soluble conductive polymer (ESpacer 300Z, Showa Denko America, Inc.)[69]. The EBL writing was performed with a voltage of 125 kV, 120 μm aperture, and 10 nA writing current. Proximity error correction was also implemented with a base dose time of 0.03 μs/dot (which corresponds to a dosage of 300 μC/cm²). The exposed photo-resist was developed by subsequently immersing the sample into water, ZED-N50 (ZEP developer), methyl isobutyl ketone, and isopropanol alcohol for 1 min each. Reactive ion etching was performed with a gas mixture of CHF₃:CF₄ (3:1) with respective flow rates of 45 sccm and 15 sccm, pressure of 10 mTorr, and RF power of 200 W. The etching rate was ~80 nm/min. The etching was done in three cycles of 5 mins with cooldown breaks of several minutes in between. After completing the etching step, the sample was soaked in N-methyl-2-pyrrolidone overnight to remove the residual ZEP resist mask. After optical characterization of the metalens in the amorphous (as-deposited) state, the sample was transitioned to the crystalline state by hot-plate annealing at 250 °C for 30 min. The annealing was conducted in a glovebox filled with an ultra-high purity argon atmosphere.

**Metasurface characterization.** The metalens sample was positioned on a three-axis translation stage and illuminated from the substrate side with a collimated 5.2-μm-wavelength laser beam (Daylight Solutions Inc., 21052-MHF-030-D00149). The focal spot produced by the metalens was magnified with a custom-made microscope assembly (henceforth termed as magnifier), consisting of lens 1 (C037TME-E, Thorlabs Inc.) and lens 2 (LA8281-E, Thorlabs Inc.). The magnified image of the focal spot was captured by a liquid nitrogen cooled InSb FPA with 320 × 256 pixels (Santa Barbara Infrared, Inc.). Magnification of the microscope assembly was calibrated to be (120 ± 3) with a USAF resolution chart. During focusing efficiency characterization, we measured optical powers of the beam passing through a reference sample (CaF₂ substrate with a deposited square gold aperture of same size as the metalens) and the metalens sample. The focusing efficiency is defined as the ratio of the power concentrated at the focal spot (within

a radius of $5\lambda_0$) over the power transmitted through the reference sample. In the metalens imaging test, we illuminated the object with a converging laser beam by placing a Si lens (LA8281-E, Thorlabs Inc.) in front of the object. A pair of singe-side polished Si wafers were inserted into the beam path as diffusers to reduce spatial coherence of the illumination beam and suppress speckles.

**Device modeling**. The full-wave meta-atom simulations were carried out with a frequency domain solver in the commercial software package CST Microwave Studio. For each meta-atom, unit cell boundary conditions were employed at both negative and positive $x$ and $y$ directions, whereas open boundary conditions were set along the $z$ axis. Each meta-atom was illuminated from the substrate side with an $x$-polarized plane wave pointing towards the positive $z$ direction. Focusing characteristics of the metalens were modeled following the Kirchhoff diffraction integral using a home-made Matlab code. The model starts with computing the Huygens point spread function of the optical system. Diffraction of the wavefront through space is given by the interference or coherent sum of the wavefronts from the Huygens' sources. The intensity at each point on the image plane is the square of the resulting complex amplitude sum.

## Data availability
The data that support the findings of this study are available from the corresponding authors upon reasonable request.

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

## Acknowledgements

The authors acknowledge X. Qiu for help with optical device modeling. This work was funded by Defense Advanced Research Projects Agency Defense Sciences Office (DSO) Program: EXTREME Optics and Imaging (EXTREME) under Agreement no. HR00111720029. The authors also acknowledge characterization facility support provided by the Materials Research Laboratory at Massachusetts Institute of Technology (MIT), as well as fabrication facility support by the Microsystems Technology Laboratories at MIT and Harvard University Center for Nanoscale Systems. The views, opinions and/or findings expressed are those of the authors and should not be interpreted as representing the official views or policies of the Department of Defense or the US Government. Distribution statement: Approved for public release. Distribution is unlimited. This material is based upon work supported by the Under Secretary of Defense for Research and Engineering under Air Force Contract No. FA8702-15-D-0001. Any opinions, findings, conclusions or recommendations expressed in this material are those of the author(s) and do not necessarily reflect the views of the Under Secretary of Defense for Research and Engineering. © 2021 Massachusetts Institute of Technology.

## Author contributions

M.Y.S. fabricated the metalens. S.A., T.G., and C.F. designed and modeled the devices. M.Y.S. and T.G. performed device characterizations. T.G., M.Y.S., and S.A. analyzed the experimental data. M.K. prepared the target materials. Y.Z., P.S., C.R., and Q.D. contributed to device fabrication. F.Y. contributed to optical testing. V.L., J.B.C., C.M.R., and C.R.-B. performed material characterizations. M.Y.S., T.G., and J.H. drafted the manuscript. J.H., T.G., H.Z., K.R., and A.A. supervised and coordinated the research. All authors contributed to revising the manuscript and technical discussions.

## Competing interests

The authors declare no competing interests.
