## [Peer Review File · Nature Communications]

Reviewers' Comments:

Reviewer #1:

Remarks to the Author:

The manuscript under review "Reconfigurable all-dielectric metalens with diffraction-limited performance" by Shalaginov et al. reports a thorough analysis and experimental proof of a dynamic all-dielectric metasurface using the unique features of GSST in the mid-IR wavelength range. The authors demonstrated an efficient switchable metalens, necessary for aberration-free and multi-depth imaging, by leveraging a generic design principle and introducing a customized FOM. Such a time-efficient approach provides a rich library of conventional meta-atoms tailoring the optical wavefront at will, with rather high performance, in the two extreme states of GSST.

Overall, this very well written manuscript with rigorous theoretical analysis and experimental proof-of-concept is a timely contribution merging two rapidly growing topics in nanophotonics, i.e., dielectric metasurface and the dynamically tunable phase-change chalcogenides. As this work can be extended to real-time tunable metasurface platforms exploiting O-PCMs for on-demand reconfigurable functionalities, I believe that this manuscript will stimulate further research and open a new path for adaptive flat optics. I thus can clearly recommend it for publication in Nature Communications after addressing the following open questions and comments.

- Can the authors elaborate the operational bandwidth of the metalens? Providing Strehl ratio as a function of wavelength is instructive.
- Besides metals, graphene, and doped silicon as compelling platforms for the conversion process, more recently ITO, as a transparent conductive oxide, microheater was used to control the state of PCMs selectively and reversibly. I think it is worth to add this to other discussed platforms.
- Polarization-insensitivity is an important feature of most optical devices. It is worth to discuss the potential of the proposed generic design principle in finding such meta-structures.
- More recently, dynamic hybrid metal-dielectric metasurfaces incorporating phase-change materials was introduced as a promising candidate for beam forming applications [arXiv:2008.03905]. The experimental demonstration reveals their potentials for shrinking the overall size of flat optical devices by leveraging pronounced plasmonic-photonic modes. Due to the moderate quality-factor of their meta-atoms, such configurations enable light manipulation in the deep subwavelength regime, which is worth to be discussed.
- How the proposed designs can be modified for realization higher NA metalenses?
- It is instructive if the authors comment why they opted operational wavelength of 5.2 μm .
- Does increase in deposition time affect the state of the as-deposited GSST? Also, it would be better to report the refractive index of GSST on the CaF₂ substrate.
- The authors mentioned that before resist coating, they treat the surface of GSST with oxygen plasma. Due to the reactive nature of PCMs, how they can be sure this step does not oxidize the material?
- Adding AFM images to show the surface morphology of amorphous and crystalline GSST could be helpful.
- (a) and (b) were not addressed in the caption of Figs. 1 and 3.
- "active" contrasts with "passive" not "static", which might be misleading. In my view, "dynamic", "tunable", or "reconfigurable" are more common in the field.
- "e.g." in line 62 page 2 should be revised as "e.g.,".

Reviewer #2:

Remarks to the Author:

Shalaginov et. al. demonstrate an all-dielectric active metalens using phase-changed material GSST. The focal length can be actively changed via thermally modulating the phase state of GSST. The authors also demonstrate a diffraction-limited imaging application at a wavelength of 5.2 micrometers. I agree that tunable metasurface is a prospective research topic, especially for the

application areas of reconfigurable focusing and beam steering. Although the use of phase-changed material for the realization of active metasurface has been widely explored, the utilization of GSST is still novel to me. However, there are some issues that have to be properly addressed in this manuscript. Detailed comments are listed below:

1. I agree that the optical efficiency of demonstrated metalens is restricted by the small number of phase discretization and limited transmittance of unit structures. However, it would be extremely challenging to overcome those issues. For example, as the number of phase discretization increased, the optimization process for unit structure can be very time-consuming, which is not efficient for practical applications. The transmittance of unit structures can be hardly improved if resonances are introduced for large phase shifts. For benefitting future applications, the authors should provide more discussions on these issues.
2. Is the demonstrated metalens polarization-independent? Because the H-shaped structure is used, I don't think the optical response is polarization-independent. If so, why the linear polarizer is absent in the optical system?
3. The authors claim that switching can be achieved by using on-chip metal micro-heaters. Is it compatible for the transmissive metasurface? If not, other approaches have to be provided and discussed.
4. Following the previous question, once the size of metalens increased, would the uniformity of current flow influence the focusing performance?
5. When the tunable metalens is phase changed to the crystalline state, how to switch the state back to amorphous?
6. It seems like the temperature of metalens has to be maintained at a certain level to ensure the survival of the crystalline state. If so, how did the authors maintain the temperature of metalens while optical measurement?

Reviewer #3:

Remarks to the Author:

Shalaginov et al present a method to design programmable metasurfaces that produce arbitrary phase profiles. They demonstrate the performance of their device using the $\text{Ge}_2\text{Sb}_2\text{Se}_4\text{Te}_1$ (GSST) PCM. The idea is based on 16 different meta-atom configurations that can be combined in different ways to optimise a cost-function that accounts for both efficiency and phase error.

Overall, I think this paper is potentially publishable in Nature Comms, and I would like the authors to consider the following points in any revised manuscript.

- Why do the authors insist on using the GSST material for this demonstration? Line 106 states that GSST is exceptionally transparent at 5.2 μm but figure 5 of reference 44 seems to show that $\text{Ge}_2\text{Sb}_2\text{Te}_5$ is more transparent. Therefore, I recommend the authors also apply their method to more common PCMs, such as those along the $\text{GeTe-Sb}_2\text{Te}_3$ pseudo-binary phase diagram. At 5.2 μm these material are also somewhat transparent [1,2]. Indeed, I think the paper will be more impactful if a comparison against $\text{GeTe-Sb}_2\text{Te}_3$ materials is included in the manuscript.
- What is the reflectivity and absorption for the results shown in Figure 4? At the moment Figure 4 shows the intensity of the focussed spot in the crystalline and amorphous states but I assume that some light is reflected or absorbed by the structure. Please present the reflectivity and transmittance of the lens.
- For the results shown in figure 4, if absorbance is significant, then there should be some heating effect. How does this heating affect the stability of the focus and the lifetime of the programmed lens?
- It is stated that a 5.2 μm collimated laser beam is focussed by the metasurface. What was the

area of the metasurface and spot size of the collimated laser?

- Lines 279–298 compare GSST with GST, but it is not compared with other emerging O-PCMs, such as Sb₂Te₃[3]. Please include this in the discussion.

- Line 281 mentions the possibility of electrically switching this material. Considering the IR reflectivity of conductors, what type of device configuration would allow electrical switching?

[1] A.-K. U. Michel, M. Wuttig, and T. Taubner. Design parameters for phase-change materials for nanos- tructure resonance tuning. *Adv. Opt. Mat.*, 5(18), 2017.

[2] L. T. Chew, W. Dong, L. Liu, X. Zhou, J. Behera, H. Liu, K. V. Sreekanth, L. Mao, T. Cao, J. Yang, and R. E. Simpson. Chalcogenide active photonics. *Proc.SPIE*, 10345:10345 – 10345 – 9, 2017.

[3] W. Dong, H. Liu, J. K. Behera, L. Lu, R. J. H. Ng, K. V. Sreekanth, X. Zhou, J. K. Yang, and R. E. Simpson. Wide band gap phase change material tuned visible photonics. *Advanced Functional Materials*, 6:1806181, 2019.

Response Letter

Dear Editor,

Thank you for handling our manuscript and for providing the opportunity to address the reviewers' concerns. We are thankful to the referees for a thorough review of our manuscript and their valuable comments. We have modified the manuscript following the reviewers' suggestions. Please see below our point-by-point response. The changes made to the manuscript are redlined.

Reviewer #1

Comment 1: Can the authors elaborate the operational bandwidth of the metalens? Providing Strehl ratio as a function of wavelength is instructive.

Our response: We thank the reviewer for his/her insightful comments. Firstly, we would like to note that our current design is not targeted for broadband operation, as detailed below. To study the operational bandwidth, we performed additional simulations of the meta-atom responses (phase and amplitude) over a range of wavelengths and calculated the corresponding Strehl ratios of the reconfigurable metasurface using the diffraction integral model. As shown in Fig. R1, the results show that the diffraction-limited bandwidth (Strehl ratio > 0.8) of our metalens is about 80 nm and 100 nm for amorphous and crystalline states, respectively. This is in a good agreement with the dispersion behavior of an ideal flat lens designed with the same center wavelength (5.2 μm) but without the wavelength-dependent phase/amplitude variance at the meta-atom level.

Fig. R1. Simulated Strehl ratio of the GSST metalens compared to an ideal lens designed at the same center wavelength (5.2 μm). The wavelength range spans from 5.1 to 5.3 μm (corresponding to frequencies of 56.6 and 58.83 THz).

Here we provide more details on the wavelength-dependent responses of our meta-atoms. In principle, the designed meta-atoms can be considered as Huygens' sources, and each individual meta-atom is represented by the combination of electric and magnetic dipoles that follow a Lorentzian frequency dependence with different resonant positions. The phase shift tuning at the specific state (amorphous or crystalline) is achieved by engineering meta-atom shapes, while the phase shift tuning between different states (amorphous to crystalline or inversely) is realized by varying dipole resonant frequencies with respect to various material indices. This indicates the narrow-band nature of the Huygens-type metasurfaces. As shown in Fig. R2, we simulated the amplitude and phase responses of the 16 designed meta-atoms over a wide spectrum (4 μm to 7.5 μm), then calculated average amplitudes for the 16 meta-atoms (Fig. R2a). Afterwards, we evaluated the average phase differences (in degrees) at each frequency point comparing to the "standard" phase shift (2π evenly divided by 4 levels) at working frequency of 5.2 μm (57.7 THz). The average phase errors are shown in Fig. R2b. As expected, the phase errors increase when operating frequency deviates from the designed working frequency.

Fig. R2. (a) Average amplitudes of the 16 meta-atoms in the 40 to 75 THz frequency range. (b) Average phase errors of the 16 meta-atoms with respect to the “standard” phase shift at working frequency. The results for amorphous state are shown in blue color, while the results for crystalline state are shown in red. The working frequency (57.7 THz, 5.2 μm) is marked with a red dashed line.

Changes in the manuscript: In the main text, we added a sentence (Results, page 7): “Additionally, we analyzed the metalens’ diffraction-limited bandwidths (i.e., wavelength range over which Strehl ratios exceed 0.8), which are approximately 80 nm and 100 nm for amorphous and crystalline states, respectively. The operational bandwidths are in good agreement with the dispersion behavior of an ideal flat lens of the same configuration (Supplementary Note 5).”

In the Supplementary, we created a new note “Supplementary Note 5 - Metalens’ bandwidth” to (pages 14-15) by adding the figures R1, R2 and the following text:

“To study the operational bandwidth, we performed additional simulations of the meta-atom responses (phase and amplitude) over a range of wavelengths and calculated the corresponding Strehl ratios of the reconfigurable metasurface using the diffraction integral model. As shown in

Supplementary Fig. 6, the results show that the diffraction-limited bandwidth (Strehl ratio > 0.8) of our metalens is about 80 nm and 100 nm for amorphous and crystalline states, respectively. This is in a good agreement with the dispersion behavior of an ideal flat lens designed with the same center wavelength (5.2 μm) but without the wavelength-dependent phase/amplitude variance at the meta-atom level.

Here we provide more details on the wavelength-dependent responses of our meta-atoms. In principle, the designed meta-atoms can be considered as Huygens' sources, and each individual meta-atom is represented by the combination of electric and magnetic dipoles that follow a Lorentzian frequency dependence with different resonant positions. The phase shift tuning at the specific state (amorphous or crystalline) is achieved by engineering meta-atom shapes, while the phase shift tuning between different states (amorphous to crystalline or inversely) is realized by varying dipole resonant frequencies with respect to various material indices. This indicates the narrow-band nature of the Huygens type metasurfaces. As shown in Fig. S7, we simulated the amplitude and phase responses of the 16 designed meta-atoms over a wide spectrum (4 μm to 7.5 μm), then calculated average amplitudes for the 16 meta-atoms (Fig. S7a). Afterwards, we evaluated the average phase differences (in degrees) at each frequency point comparing to the "standard" phase shift (2π evenly divided by 4 levels) at working frequency of 5.2 μm (57.7 THz). The average phase errors are shown in Fig. S7b. As expected, the phase errors increase when operating frequency deviates from the designed working frequency."

Comment 2: Besides metals, graphene, and doped silicon as compelling platforms for the conversion process, more recently ITO, as a transparent conductive oxide, microheater was used to control the state of PCMs selectively and reversibly. I think it is worth to add this to other discussed platforms.

Our response: Following the reviewer's suggestion, we have added a reference and modified the sentence about the available micro-heating approaches. Modified sentence (Discussion, page 10, second paragraph): "Additionally, reversible switching of GSST and other phase change materials using transparent graphene⁵⁸, indium-tin oxide[Kato, K., Kuwahara, M., Kawashima, H., Tsuruoka, T. & Tsuda, H. Current-driven phase-change optical gate switch using indium-tin-oxide heater. Appl. Phys. Express 10, 072201 (2017); Taghinejad, H. et al. ITO-Based Microheaters for Reversible Multi-Stage Switching of Phase-Change Materials: Towards Miniaturized Beyond-Binary Reconfigurable Integrated Photonics. arxiv: 2003.04097 (2020)], and doped Si⁶⁰ heaters have also been validated.

Comment 3: Polarization-insensitivity is an important feature of most optical devices. It is worth to discuss the potential of the proposed generic design principle in finding such meta-structures.

Our response: The most straightforward approach to design meta-atoms with polarization-independent performance is to find the unit-cell cross-section geometries with a 4-fold symmetry. As an example, we selected 16 meta-atom designs with randomly-generated 4-fold symmetrical shapes (Fig. R3). The selected meta-atoms can provide full 2π phase coverage and enable the required 16 phase-delay responses in amorphous and crystalline states. Similar to the original designs shown in Fig. 2k of the main manuscript, this meta-atom set can be employed to generate arbitrary wavefronts in both states. The simulated values of phase shifts and transmittance of the polarization-independent meta-atoms are listed in Table R1. The average transmitted field amplitudes of 4-fold symmetric patterns are 0.65 and 0.6 in A-state and

C-state, respectively, which can be further improved by increasing design degrees of freedom and using advanced optimization methods [An, S. et al. A Deep Learning Approach for Objective-Driven All-Dielectric Metasurface Design. ACS Photonics 6, 3196–3207 (2019)].

Fig. R3. (a) Schematic top-view of the 4-fold symmetry 2-bit meta-atom designs; (b) simulated phase and amplitude of the 16 meta-atoms in amorphous state; (c) simulated phase and amplitude of the 16 meta-atoms in crystalline state.

Table R1. Phase delays and transmittances of the polarization-independent meta-atoms in amorphous and crystalline states of GSST.

cell	amorphous		crystalline	
	phase, °	T, %	phase, °	T, %
1	6.9	16.21	3.8	25.43
2	0.1	24.71	89.7	48.60
3	0.7	17.14	179.3	46.76
4	1.0	24.78	271.9	20.21
5	89.7	65.41	359.5	72.20
6	90.0	79.98	89.6	16.61
7	86.6	86.73	176.4	21.01
8	84.9	86.39	263.9	25.04
9	180.6	57.92	359.7	27.21
10	182.9	41.67	91.0	18.69
11	159.2	93.76	151.1	93.99
12	170.6	50.63	342.5	16.06
13	267.9	19.80	5.2	18.96
14	272.0	23.92	88.0	74.01
15	316.0	33.82	132.1	84.77
16	361.0	24.78	271.9	20.21

Changes in the manuscript: In the in the Supplementary (pages 16-17), we added a note “Supplementary Note 6 - Polarization-insensitive reconfigurable meta-atoms” containing the geometries polarization-independent meta-atoms and their properties. The section includes Fig. R3, Table R1 and the following text: “The most straightforward approach to design meta-atoms with polarization-independent performance is to find the unit-cell cross-section geometries with a 4-fold symmetry. As an example, we selected 16 meta-atom designs with randomly-generated 4-fold symmetrical shapes (Supplementary **Error! Reference source not found.**). The selected meta-atoms can provide full 2π phase coverage and enable the required 16 phase-delay responses in amorphous and crystalline states. Similar to the original designs shown in Fig. 2k of the main manuscript, this meta-atom set can be employed to generate arbitrary wavefronts in both states. The simulated values of phase shifts and transmittance of the polarization-independent meta-atoms are listed in Supplementary **Error! Reference source not found.** The average transmitted field amplitudes of 4-fold symmetric patterns are 0.65 and 0.6 in A-state and C-state, respectively, which can be further improved by increasing design degrees of freedom and using advanced optimization methods [An, S. et al. A Deep Learning Approach for Objective-Driven All-Dielectric Metasurface Design. *ACS Photonics* 6, 3196–3207 (2019)].”

Comment 4: More recently, dynamic hybrid metal-dielectric metasurfaces incorporating phase-change materials was introduced as a promising candidate for beam forming applications [arXiv:2008.03905]. The experimental demonstration reveals their potentials for shrinking the overall size of flat optical devices by leveraging pronounced plasmonic-photonic modes. Due to the moderate quality-factor of their meta-atoms, such configurations enable light manipulation in the deep subwavelength regime, which is worth to be discussed.

Our response: We added the suggested reference in a sentence of the Introduction (page 2): “Many studies have achieved amplitude or spectral tailoring of light via metastructures made of these materials^{27–37}”.

37. Abdollahramezani, S. *et al.* Dynamic hybrid metasurfaces. *arXiv: 2008.03905* 1–14 (2020).

Comment 5: How the proposed designs can be modified for realization higher NA metalenses?

Our response: In our prior work, we have implemented a 0.7-NA aspheric metalens made of PbTe meta-atoms [Zhang, L. et al. Ultra-thin high-efficiency mid-infrared transmissive Huygens meta-optics. *Nat. Commun.* 9, 1481 (2018)]. Since our approach allows binary switching between arbitrary phase profiles, by using the same meta-atom library and high-NA-metalens phase profiles it should be straight forward to realize a varifocal metalens with higher NAs. We also showed in the previous paper that the ultra-thin profile of the Huygens meta-atoms presents additional advantages in terms of optical efficiency thanks to reduced shadowing.

Comment 6: It is instructive if the authors comment why they opted operational wavelength of 5.2 μm .

Our response: We have selected the operational wavelength at 5.2 μm due to the available laser source in our laboratory (we have an external cavity tunable laser at 5.1-5.4 μm) and low material absorption at wavelengths $> 3 \mu\text{m}$.

Comment 7: Does increase in deposition time affect the state of the as-deposited GSST? Also, it would be better to report the refractive index of GSST on the CaF₂ substrate.

Our response: We don't observe any significant effect of deposition time on the material properties of as-deposited GSST. Throughout the deposition process, the substrate temperature was kept below 40°C, measured by a thermocouple. We also confirmed amorphous nature of the as-deposited films using Raman and X-ray diffraction. The GSST n&k data are provided in our previous work [Zhang, Y. et al. Broadband transparent optical phase change materials for high-performance nonvolatile photonics. Nat. Commun. 10, 1–9 (2019)] and has been already cited in the current manuscript.

Comment 8: The authors mentioned that before resist coating, they treat the surface of GSST with oxygen plasma. Due to the reactive nature of PCMs, how they can be sure this step does not oxidize the material?

Our response: We expect that the material oxidation depth is limited to a few nanometers on the surface even after extensive (5 min and above) oxygen plasma treatment, according to measurements performed on plasma treated GST [Golovchak, R. et al. Oxygen incorporation into GST phase-change memory matrix. Appl. Surf. Sci. 332, 533–541 (2015)]. Data pertaining to GSST is not available, although we anticipate the two materials will exhibit identical behavior given their chemical similarity.

Comment 9: Adding AFM images to show the surface morphology of amorphous and crystalline GSST could be helpful.

Our response: We included the AFM images of the GSST films in both amorphous and crystalline states. The material maintains low RMS surface roughness of < 0.2 nm (amorphous) and 2.2 nm (crystalline). This minimal level of roughness has negligible impact on optical responses of the meta-atoms. Similar AFM results are provided in our prior work [Zhang, Y. et al. Broadband transparent optical phase change materials for high-performance nonvolatile photonics. Nat. Commun. 10, 1–9 (2019)].

Fig. R4. Surface topology of the GSST film: (a) amorphous (as-deposited), (b) thermally crystallized (annealed at 300°C for 30 mins).

Comment 10: (a) and (b) were not addressed in the caption of Figs. 1 and 3.

Our response: We have removed labels (a) and (b) from the Figs. 1 and 3. Please refer to the updated figures.

Comment 11: “active” contrasts with “passive” not “static”, which might be misleading. In my view, “dynamic”, “tunable”, or “reconfigurable” are more common in the field.

Our response: We would like to keep ‘active metasurface’ as a well-established term. In other cases, we substituted the word ‘active’ with the adjectives suggested by the reviewer.

Comment 12: “e.g.” in line 62 page 2 should be revised as “e.g.,”.

Our response: We corrected this typo.

Reviewer #2

Comment 1: I agree that the optical efficiency of demonstrated metalens is restricted by the small number of phase discretization and limited transmittance of unit structures. However, it would be extremely challenging to overcome those issues. For example, as the number of phase discretization increased, the optimization process for unit structure can be very time-consuming, which is not efficient for practical applications. The transmittance of unit structures can be hardly improved if resonances are introduced for large phase shifts. For benefitting future applications, the authors should provide more discussions on these issues.

Our response: We thank the reviewer for this important comment and we do agree that indeed building multifunctional metasurface designs with higher levels of discretization and high optical efficiencies is a challenging problem. To overcome this challenge, we have developed several approaches for generating freeform meta-atom geometries with the desired optical responses as well as for increasing their efficiency by a deep-learning-based design scheme [Shalaginov, M. Y. et al. Design for quality: reconfigurable flat optics based on active metasurfaces. *Nanophotonics* 9, 3505–3534 (2020)]. For instance, recently we leveraged generative-adversarial neural networks (GANs) for producing a library of 64 GSST meta-atoms for achieving 8-level phase discretization with reduced phase errors across the two states (Fig. R5). We are summarizing these results into an upcoming publication.

Fig. R5. Exemplary library of 64 GSST meta-atoms for enabling 8-phase-level binary reconfigurable metasurfaces at 5.2- μm -wavelength. Horizontal and vertical directions in the table indicate meta-atom phase-shifts (from -180° to $+180^\circ$) in amorphous (A-state) and crystalline (C-state) states, respectively.

Comment 2: Is the demonstrated metalens polarization-independent? Because the H-shaped structure is used, I don't think the optical response is polarization-independent. If so, why the linear polarizer is absent in the optical system?

Our response: Yes, the demonstrated metalens is not polarization-independent. The incident beam comes directly from the external cavity laser source, which is linearly polarized. We have

experimentally verified the laser polarization with a polarizer. We would like to mention that we can also generate a library of polarization-independent meta-atoms, please see the response to Comment 3 from Reviewer 1.

Comment 3: The authors claim that switching can be achieved by using on-chip metal microheaters. Is it compatible for the transmissive metasurface? If not, other approaches have to be provided and discussed.

Our response: We agree that development of transparent microheaters is of vital importance for the next generation of reconfigurable transmissive metasurfaces based on phase-change materials. In the Introduction section (page 10) we have pointed out at several works on transparent microheaters: “Additionally, reversible switching of GSST and other phase change materials using transparent graphene⁵⁵, indium-tin oxide^{56,57}, and doped Si⁵⁸ heaters have also been validated.” We have successfully implemented graphene and doped Si heaters for reversible GSST switching in our group.

Comment 4: Following the previous question, once the size of metalens increased, would the uniformity of current flow influence the focusing performance?

Our response: Uniformal heating is essential for precise control of each meta-atom optical response, which directly influences the overall focusing performance. The uniformity of temperature distribution can be achieved by optimizing the shape of a heating element as we have experimentally validated in [Zhang, Y. et al. Electrically Reconfigurable Nonvolatile Metasurface Using Low-Loss Optical Phase Change Material. arxiv: 2008.06659 (2020)], an approach applicable to all heater architectures. Alternatively, we have also shown that the doping profile in Si heaters can be engineered to maximize thermal uniformity. Fig. R6 shows an exemplary Si heater design following this approach.

Fig. R6. Numerical simulations of a fan-shaped doped-Si heater. (a) Boron doping concentration ranging from $4 \times 10^{17} \text{ cm}^{-3}$ to $4 \times 10^{19} \text{ cm}^{-3}$. Color bar shows concentration values on a logarithmic scale. Electrodes on the heater sides are colored in black. (b) Temperature distribution over the fan-shaped Si heater. Average temperature is 610 K, at which GSST thermally crystallizes; temperature contrast is $< 7\%$.

Comment 5: When the tunable metalens is phase changed to the crystalline state, how to switch the state back to amorphous?

Our response: Re-amorphization of GSST is typically achieved by a melt-quenching process, i.e., heating the material above the melting temperature followed by rapid cooling. Melt-quenching can be triggered by 10- μs -long electrical pulses as we have demonstrated experimentally [Zhang, Y. et al. Electrically Reconfigurable Nonvolatile Metasurface Using

Low-Loss Optical Phase Change Material. arxiv: 2008.06659 (2020), Ríos, C. et al. Multi-level Electro-thermal Switching of Optical Phase-Change Materials Using Graphene. arXiv: 2007.07944 (2020)].

Comment 6: It seems like the temperature of metalens has to be maintained at a certain level to ensure the survival of the crystalline state. If so, how did the authors maintain the temperature of metalens while optical measurement?

Our response: GSST is a non-volatile phase-change alloy, i.e., the material preserves its structural state without any external stimulus. Hence, there is no need to keep it at elevated temperatures. Non-volatility is actually one of the key advantages of our material platform, which potentially allows to significantly reduce device power consumption. We have clarify this attribute in the main text.

Changes in the manuscript (Results, page 3): “We selected $\text{Ge}_2\text{Sb}_2\text{Se}_4\text{Te}_1$ (GSST) as a **non-volatile** O-PCM to construct the metasurface operating at the wavelength $\lambda_0 = 5.2 \mu\text{m}$.”

Reviewer #3

Comment 1: Why do the authors insist on using the GSST material for this demonstration? Line 106 states that GSST is exceptionally transparent at 5.2 μm but figure 5 of reference 44 seems to show that $\text{Ge}_2\text{Sb}_2\text{Te}_5$ is more transparent. Therefore, I recommend the authors also apply their method to more common PCMs, such as those along the $\text{GeTe-Sb}_2\text{Te}_3$ pseudo-binary phase diagram. At 5.2 μm these material are also somewhat transparent [1,2]. Indeed, I think the paper will be more impactful if a comparison against $\text{GeTe-Sb}_2\text{Te}_3$ materials is included in the manuscript.

Our response: We thank the reviewer for the detailed review. In the Figure 5 of reference 44 [Zhang, Y. et al. Broadband transparent optical phase change materials for high-performance nonvolatile photonics. Nat. Commun. 10, 1–9 (2019)], the extinction coefficient for crystalline GSST (GSS4T1) at 5.2- μm -wavelength is $k = 0.04$, which is 50 times smaller than k for crystalline GST-225 ($k = 2$ at 5.2 μm). For our metalens design, we selected the O-PCM with the minimal losses in both amorphous and crystalline states. The crystalline state k of GSST is also only half of that of GST-326 ($k = 0.08$) [Data from: Michel, A.-K. U. et al. Using Low-Loss Phase-Change Materials for Mid-Infrared Antenna Resonance Tuning. Nano Lett. 13, 3470–3475 (2013)], a PCM composition with optimal transparency in the $\text{GeTe-Sb}_2\text{Te}_3$ pseudo-binary family. More importantly, $\text{GeTe-Sb}_2\text{Te}_3$ alloys mandate much higher cooling rates to ensure complete re-amorphization during melt-quenching, which sets a limit on the meta-surface thickness of approximately 100 nm or less [Ruiz de Galarreta, C. et al. Reconfigurable multilevel control of hybrid all-dielectric phase-change metasurfaces. Optica 7, 476 (2020)]. With such material thickness it is not possible to achieve a full 2π phase coverage in all-dielectric meta-atoms in the IR band.

Comment 2: What is the reflectivity and absorption for the results shown in Figure 4? At the moment Figure 4 shows the intensity of the focussed spot in the crystalline and amorphous states but I assume that some light is reflected or absorbed by the structure. Please present the reflectivity and transmittance of the lens.

Our response: We simulated transmittance (T), reflectance (R), and absorptance (A) values of all the 16 meta-atoms, and the results are illustrated in Fig. R7. Therefore, we see that the reflected and absorbed power is a relatively small fraction compared to transmitted light.

Fig. R7. Simulated transmittances (blue), reflectances (red) and absorptances (yellow) of the 16 meta-atoms. The T, R, A values averaged over all meta-atoms at the operational wavelength of $5.2 \mu\text{m}$ in amorphous (crystalline) state are $T_A = 66\%$ ($T_C = 70\%$); $R_A = 20\%$ ($R_C = 17\%$); $A_A = 14\%$ ($A_C = 13\%$).

Comment 3: For the results shown in figure 4, if absorbance is significant, then there should be some heating effect. How does this heating affect the stability of the focus and the lifetime of the programmed lens?

Our response: We have numerically studied the heating effect due to the absorption of the incident laser light by a metasurface. To estimate the upper boundary of the temperature increase, we considered a meta-atom #9 in amorphous state (Fig. R7), which is expected to experience the highest absorbance among the other meta-atoms. By using COMSOL Multiphysics, we have verified that under illumination of laser light with the intensity of 34 kW/m^2 , which resembles the experimental conditions, the heating effect is negligible with a temperature rise of $< 0.1 \text{ K}$.

Fig. R8. Simulated change in temperature caused by a meta-surface unit-cell absorbing the incident laser beam.

Comment 4: It is stated that a 5.2 μm collimated laser beam is focussed by the metasurface. What was the area of the metasurface and spot size of the collimated laer?

Our response: The metasurface area is $1.5 \times 1.5 \text{ mm}^2$, laser beam diameter is approximately 3 mm.

Comment 5: Lines 279—298 compare GSST with GST, but it is not compared with other emerging O-PCMs, such as a Sb_2Te_3 [3]. Please include this in the discussion.

Our response: In the Discussion section (page 10) we added the following sentence. “It is worth mentioning that our general reconfigurable metasurface design principle can be also implemented with other O-PCMs, such as Sb_2S_3 [Dong, W. et al. Wide Bandgap Phase Change Material Tuned Visible Photonics. *Adv. Funct. Mater.* 29, 1806181 (2019).], Sb_2Se_3 [Delaney, M., Zeimpekis, I., Lawson, D., Hewak, D. W. & Muskens, O. L. A New Family of Ultralow Loss Reversible Phase - Change Materials for Photonic Integrated Circuits: Sb_2S_3 and Sb_2Se_3 . *Adv. Funct. Mater.* 30, 2002447 (2020)], and $\text{Ge}_3\text{Sb}_2\text{Te}_6$ [Michel, A.-K. U., Wuttig, M. & Taubner, T. Design Parameters for Phase-Change Materials for Nanostructure Resonance Tuning. *Adv. Opt. Mater.* 5, 1700261 (2017)].”

Comment 6: Line 281 mentions the possibility of electrically switching this material. Considering the IR reflectivity of conductors, what type of device configuration would allow electrical switching?

Our response: Please see the response to the Comment 3 from Reviewer 2.

Reviewers' Comments:

Reviewer #1:

Remarks to the Author:

In this revised version, the authors have answered all comments raised by the referees in a satisfactory manner. The changes and the added materials remarkably improved the manuscript, so I recommend publication of this work.

Reviewer #2:

Remarks to the Author:

The authors did a great job on the revision. I agree that this manuscript can be accepted for publication without further modification.

Reviewer #3:

Remarks to the Author:

The Authors have adequately addressed all of my queries. I can recommend publication pretty much as is.